# The Matrix KV Storage System Based on NVM Devices

**DOI:** 10.3390/mi10050346

**Published:** 2019-05-27

**Authors:** Tao Cai, Fuli Chen, Qingjian He, Dejiao Niu, Jie Wang

**Affiliations:** School of Computer Science and Communication Engineering, Jiangsu University, Zhenjiang 212013, China; 13027520312@163.com (F.C.); he_qingjian@outlook.com (Q.H.); wangjie287@live.com (J.W.)

**Keywords:** key value storage system, nonvolatile memory, key value pairs management, I/O system software stack

## Abstract

The storage device based on Nonvolatile Memory (NVM devices) has high read/write speed and embedded processor. It is a useful way to improve the efficiency of Key-Value (KV) application. However it still has some limitations such as limited capacity, poorer computing power compared with CPU, and complex I/O system software. Thus it is not an effective way to construct KV storage system with NVM devices directly. We analyze the characteristics of NVM devices and demands of KV application to design the matrix KV storage system based on NVM Devices. The group collaboration management based on Bloomfilter, intragroup optimization based on competition, embedded KV management based on B+-tree, and the new interface of KV storage system are presented. Then, the embedded processor in the NVM device and CPU can be comprehensively utilized to construct a matrix KV pair management system. It can improve the storage and management efficiency of massive KV pairs, and it can also support the efficient execution of KV applications. A prototype is implemented named MKVS (the matrix KV storage system based on NVM devices) to test with YCSB (Yahoo! Cloud System Benchmark) and to compare with the current in-memory KV store. The results show that MKVS can improve the throughput by 5.98 times, and reduce the 99.7% read latency and 77.2% write latency.

## 1. Introduction

The disk and flash-based solid-state device (SSD) cannot provide high random I/O performance. The I/O performance is much lower than the speed of a CPU and the storage device is the bottleneck of computer system. Nowadays, a series of NVM (nonvolatile memory) has been developed, such as phase-change memory (PCM) [1], shared transistor technology random access memory (STT-RAM) [2], Intel’s 3D X-point [3], etc. These devices have the advantages of being byte-addressable, longer writing lifetime compared with Flash, low power consumption, and close to the I/O speed of dynamic random access memory (DRAM). Then the NVM device has high read and write speed. However the current I/O system software stack was designed for low-speed storage devices, which becomes an important factor affecting the performance of the NVM storage system. Relevant research indicates that the I/O system software stack accounts for more than 94% of the time overhead in the NVM storage system. Therefore, how to reduce the time overhead of the I/O system software stack is important to improve NVM storage system performance. Key-value (KV) store is an important application and it has been widely used in many systems, such as Dynamo at Amazon [4], Voldemort at LinkedIn [5], Cassandra at Apache [6], LevelDB at Google [7], and RocksDB at Facebook [8]. It is tailored in many data-intensive internet applications, such social networking, e-commerce, and online gaming. Currently, the NVM device is used instead of an HDD (hard disk drive) or Flash-based SSD (solid state drive). The KV application generally needs to go through the file system when accessing the storage system. However, research shows that the latency of the file system accounts for 30% of the total delay, resulting in an 85% performance loss. Therefore, how to design new KV storage systems according to the characteristics of NVM devices and the KV application is an important factor in improving the performance of the KV application.

The NVM device has the embedded processor and it can share some management task for storage system or application. However, compared with CPU, the embedded processor of NVM device has limited computing power, and it is difficult to index and management of a large number of KV pairs independently. At the same time, the capacity of a single NVM device is limited, and it is difficult to meet the demand for large-capacity KV store. Building a storage array with multiple NVM devices can achieve huge storage space, and using multiple NVM devices can effectively share the index and search tasks of KV pairs. Therefore, building storage arrays with NVM devices is an effective way to solve the problem of high-efficiency storage and management of a large number of KV pairs. However, the traditional storage array is a server-independent storage device that must be connected to the server using a network or interface. The read/write speed of the NVM device is much higher than that of the traditional storage device. It makes the network or interface become a bottleneck and affects the performance of the NVM array. Therefore, integrating multiple NVM devices into the server, constructing the server’s embedded storage array, and reducing the hardware and software overhead when accessing storage arrays are important issues.

BloomFilter is a commonly used to distribute data and access requests in a distributed system. However its computation time and space overhead are very large, it is not suitable for embedding into NVM devices, and calculates using the embedded processor in NVM devices. The performance of NVM devices has instability due to wear leveling and garbage collection. Thus, it is not a good choice to directly using BloomFilter in NVM arrays. Therefore, it is necessary to study a new distributed KV management strategy for the characteristics of NVM arrays.

The main contributions of this paper are as follows.
We comprehensively utilize the embedded processor in the NVM device and the CPU to construct a matrix KV pair management mechanism. The CPU completes the KV pair distribution task with high computational overhead, and then the embedded processor separately indexes part of the KV pair stored in the NVM device. Thereby it can improve the efficiency of mass KV storage and management, and also improve the efficiency of the KV applications.The KV pairs are distributed among multiple groups of NVM devices in the host. The multiple sets of NVM devices are used to share the storage and management tasks of the massive KV pairs. Thereby it can avoid the limited capacity of a single NVM device, realize efficient storage and management of massive KV pairs, improves the efficiency and concurrency of KV pair access, and increases the execution speed of KV applications.One NVM devices group consists of several NVM devices, and the competition of these NVM devices improves the efficiency of KV pair storage and management. Then it can avoid the impact of NVM device read/write speed instability on access KV efficiency.The KV engine is embedded in each NVM device, and the distributed KV management system in the host is built. Then the KV pair can be managed by the embedded processor of the NVM device. The exchange of a large number of KV pairs between the NVM device and the host can beThe I/O software stack for accessing KV pairs can be shorten to adapt to the characteristics of NVM arrays and KV applications. It can improve the efficiency of KV pair access and management for KV application. At the same time, several system calls are added to encapsulate the KV management operations of matrix KV storage system; this can ensure the compatibility of existing applications and avoid numerous modifications to current KV applications.

## 2. Related Works

There are many researches on how to improve the access speed of a storage system based on NVM devices. A PCIe PCM array named Onyx was implemented and tested [9]. The results showed that it could improve the performance of read and small write by ~72–120% compared with Flash-based SSD. FusionIO extended the support of file systems for atomic writing [10]. It could convert write requests in MySQL to atomic write requests by sending to NVM devices, and improved the efficiency of transactions by 70%. S. Kannan, A, used NVM devices to store checkpoints locally and remotely [11]. The pre-replication policy was designed to move the checkpoint from memory to NVM devices before the checkpoint is started. The efficiency of metadata management largely affects the I/O performance of a file system. In general, metadata are stored with the data of a file in blocks, and a small modification to the metadata leads to the update of the entire block. To take the advantage of the high efficiency of NVM devices to optimize the access performance of the metadata, Subramanya R Dulloor designed a lightweight file system called PMFS for NVDIMMs (non-volatile dual in-line memory module) [12]. It uses cache lines and a 64-byte granulated log to ensure file system consistency, while reducing the performance impact of metadata updating and balances supporting existing applications and optimizing the access performance on NVM devices. Youyou Lu designed Blurred Persistence for transactional persistent memory [13]. It could blur the volatility–persistence boundary to reduce the overhead in transaction support and improve system performance by 56.3% to 143.7%. Wei Q proposed the persistence in memory metadata management mechanism (PIMM), which reduces SSD I/O traffic by utilizing persistence and byte-addressable of NVM devices [14]. PIMM separated data from the metadata access path and stored the data on SSD at runtime and the metadata on NVM devices. PIMM is prototyped on a real NVDIMM platform. Extensive evaluation on the implemented prototype showed that it could reduce the block erase of SSD by 91% and improve I/O performance under several real workloads.

Additionally, much work has been done on the system software overhead and performance loss caused by the access interface. Swanson analyzed the hardware and software overhead of the storage system based on NVM devices [15], pointing out that the current I/O system software stack needs to be reconstructed. The proportion on the software stack for traditional architecture is 18%, while PCIe-NVM is 63%. The overhead of the software greatly prevents the NVM device from achieving the purpose of increasing bandwidth and reducing latency. This article points to the shortcomings of using the traditional block I/O interface, and proposes a primitive that batch processes multiple I/O requests to achieve atomic writes, which can reduce overhead of applications, file systems, and operating systems. Besides, direct access to NVM devices is also very popular. DEVFS also reminded that the software stack of storage system should be carefully considered when exploring the characteristics of the storage device [16]. The traditional software stack of storage requires the application to be trapped in the operating system and involved in many layers such as memory buffers, cache, file system, block layer, etc. They will undoubtedly greatly increase the access latency, thus reducing the benefits of the NVM device with high I/O speed. Researchers reminded readers that file systems account for a large proportion of software overhead, so it is important to optimize or current file systems. In the PCIe-NVM prototype system, the file system accounts for 30% of the total latency and reduces performance by 85%. Volos explored the interface optimization technology for SCM (storage class memory) [17], and proposed to use the hardware access control function to avoid the time overhead of the context switch between the kernel space and user space when accessing the file system and to spread the file system function into the application to achieve more flexibility. Although the performance of the system can be greatly improved by adjusting the existing storage I/O stack, the programming based on the fixed POSIX interface is still too cumbersome and inefficient and is not friendly enough to the programmer. In this regard, research on direct I/O has also been carried out, which allow users to interact directly with memory without modifying metafiles, while reducing the access control overhead for data at the file system level. The hardware-based file system access control was used to separate access paths for metadata and data [18,19], and a direct I/O between the user space storage devices was used to avoid metadata modification. In addition, in order to take advantage of the byte addressing of NVM devices, it is necessary to pay attention to the granularity of access and update. Mnemosyne is a lightweight access interface for NVDIMMs to solve the problem of how user programs create and manage nonvolatile memory and how to ensure data consistency [20]. The load/store instruction was used to access the NVDIMMs directly [21].

Many studies were focused on how to improve the search efficiency and access performance of KV pairs according to the characteristics of KV store. Data-intensive storage in the age of big data urgently demands a flexible and efficient KV store, in particular in the field of web services. KV store is responsible for storing large amounts of data and accessing them quickly. KV store consists of massive small files, and the saccess characteristics and proportion of each operation must be considered when designing the system. Xingbo Wu designed LSM-Trie [22], a prefix tree structure that can effectively manage metadata and reduce write time overhead of KV store. The combination of KV store and NVM was also a hot spot. The hybrid storage was praised due to the read and write performance gap between NVM devices and DRAM. For NVM, especially PCM has limited writing lifetime. Therefore, there are many researches on how to optimize NVM devices to reduce write times [23,24,25,26]. For example, Chen proposed an unordered leaf node B+ tree to reduce the write overhead caused by sorting [23]. HIKV realize the overall optimization of KV operation by using the advantages of hybrid storage and hybrid indexing [27]. In order to take the advantage of byte-addressable for the NVM device, Deukyeon redesigned the B+ tree to overcome the problem that the amount of write transmission data is inconsistent with cache line. The open source project Pmemkv is a KV database for NVM devices [28]. It uses the linked list and C++ binding of the persistent memory development kit (PMDK) libpmemobj library to implement a persistent memory-aware queue for direct memory access. NVMKV optimizes the KV store based on the internal structure of the NVM device, and implements a lightweight KV store using an FTL-sparse address space, dynamic mapping technology, and transaction consistency, while supporting a highly lock-free parallel mechanism [29] that can almost reach bare device speed. The workload analysis of cache showed that the ratio of get operation to set operation is up to 30:1 in KV store. This means that the concurrency is very important demand to the storage system for KV store. The NVM device has good parallelism. Echo [30] and NVStore [31] use MVCC for concurrency control. Chronos [32] and MICA [33] use partitioning to achieve concurrency control for hash tables. PALM is a lock-free concurrent B+ tree [34]. FPTree use HTM (hardware transactional memory) to handle the concurrency of internal nodes and use fine-grained locks for leaf nodes to access concurrently [18]. ALOHA-KV proposes counter example to prove that concurrent transactions can reduce the time overhead without conflicts, and designs the epoch-based ECC (error correcting code) mechanism to minimize the overhead caused by synchronization conflicts [35].

## 3. Challenges

● The embedded processor of NVM devices

Similar to Flash-based SSDs, NVM devices generally also equipped with embedded processors that have a certain amount of computing power while completing the internal management functions of NVM device. However, the existing KV storage system lacks a corresponding optimization strategy, and cannot utilize the embedded processor to share the KV storage system management function, which also affects the I/O performance of the KV storage system while wasting computing resources. When the efficiency of the access interface of the NVM storage system is difficult to increase rapidly, how to distribute the management functions to the NVM device in a way of efficiently managing and processing massive data in a big data system can reduce the number of accesses to the NVM device in the KV storage system and the impact of slow access interfaces on KV storage system performance.

● The limitations of the NVM device

A single NVM device has limited capacity and it is difficult to store massive KV pairs. The NVM array is an effective way to build a large-capacity storage system. Unlike traditional disks, the performance of NVM devices is instability due to wear leveling and garbage collection. The read/write speed of NVM devices will change with different amount of data stored and different period of lifetime, which increases the complexity of NVM array management.

● The low speed interface of NVM storage system

With the concurrency of multiple NVM devices, NVM devices can achieve extremely high I/O performance but the speed of interface is relatively slower than the NVM storage system. Currently, PCIe (Peripheral Component Interconnect Express), SAS (Serial Attached SCSI), and SATA (Serial ATA) are commonly used interfaces for NVM devices with a large storage capacity. Due to the limitation of interface type and protocol, the transmission speed of NVM device interface is generally slower than the read/write speed inside NVM device. It is restricted by the structure and hardware of whole computer system and is difficult to make an improvement in a short time.

● Complex I/O software stack

When reading and writing NVM devices, the KV store needs to go through system calls, file systems, page caches, block layers, etc., which not only consist of multiple data conversion, but also involve several data copies between the user space and kernel space. At the same time, there are several caches, such as page caches and caches, in NVM devices. The read and write speed of the NVM device is already close to the DRAM that built these caches, and these caches should affect efficiency of KV store. In addition, the current I/O system software stack is designed and optimized for accessing a large amount of data, and it is difficult to effectively adapt to random read and write of small data for KV pairs.

## 4. System Design

### 4.1. The Architecture of MKVS

We modified the structure of the current KV store and I/O system software stack to design a new matrix KV storage system based on NVM Devices named MKVS. The structure is shown in Figure 1.

Firstly, several layers in the current I/O system software stack are bypassed such as page cache, file system and general block layer, etc. It can shorten the I/O system software stack to reduce the time overhead of access KV pairs. The group collaboration management module based on Bloomfilter, intragroup optimization module based on competition, embedded KV management module based on B+-tree, and the new interface of MKVS are added. The group collaboration management module based on Bloomfilter is responsible for distributing the KV pairs among multiple NVM device groups by using Bloomfilter and distributing the KV application access requests into the corresponding NVM device groups to realize distributed storage and management of KV pairs. The intragroup optimization module based on competition is responsible for dealing with KV pair access requests by multiple NVM devices in the same NVM device group competitively. The embedded KV pair management based on the B+-tree module is responsible for establishing an index of KV pairs stored in the NVM device and efficiently managing these KV pairs by using an embedded processor of the NVM device. In addition, some system calls are added to the interface of MKVS so that KV applications can skip the KV database, file system, page cache, and general block layer and directly access the corresponding functions in MKVS.

MKVS distributes KV pairs among several groups of NVM devices, which can avoid the problem of limited capacity of a single NVM device and ensure the efficiency of massive KV pair storage and management. Multiple groups of NVM devices can effectively share the storage and management tasks of massive KV pairs, improve the efficiency and concurrency of KV pairs access. Then the execution performance KV applications can be improved. The Bloomfilter needs large time and space overhead; it is computed by the CPU. Consequently, it can avoid excessive consumption of the embedded processor in the NVM device and prevent a bottleneck of management efficiency for massive KV pairs. One NVM device group contains multiple NVM devices, and the efficiency of accessing and managing KV pairs can be improved by utilizing the concurrency of the several NVM devices. Then the influence of KV store efficiency can be avoided by performance instability of NVM device. Meanwhile, the time overhead of accessing KV pairs can also be reduced by the complete of several NVM devices in one NVM device group. In general, MKVS can comprehensively utilize the embedded processor in the NVM device and CPU to construct a matrix KV pair management system. The CPU completes the distribution task of KV pairs with high time overhead, and then the embedded processor maintains indexes for KV pairs stored in NVM devices, respectively. Therefore, the heterogeneous distributed system for KV management can be established to improve the storage and management efficiency of massive KV pairs, and it also can support the efficient execution of KV applications.

### 4.2. The Group Collaborative Management Based on BloomFilter

MKVS contains multiple NVM device groups. By distributing the KV pairs among these groups, the number of KV pairs that each NVM device needs to store and manage can be reduced. Thereby the efficiency of accessing KV pairs can be improved. Then how to distribute KV pairs among NVM device groups is important to affect the efficiency of MKVS. We design a group collaborative management based on BloomFilter.

BloomFilter can be used to distribute data and access requests in the distributed computing system. However, its calculation requires a lot of time and space overhead. Therefore, the CPU is used to calculate the key of the KV pairs by BloomFilter and assign the access request to the corresponding NVM device group. The Counting BloomFilter is used in group collaborative management and is based on BloomFilter. It can avoid the difficulty of deletion operation for the BloomFilter. When using the group collaborative management based on BloomFilter, the pseudocode for the KV pair insert operation is as follows.

  Collaborative_Insert (Key, Value)  Read the Key and Value of the KV pair;  Use Counting Bloomfilter to calculate the hash value of the Key;  If (There is a corresponding hash value) {    Find the corresponding NVM devices group;   Call the lookup KV interface of the NVM device;    If (There is no corresponding Key in the NVM device) {    Write the KV pair using the insert KV interface of NVM device;}  Else{   Return a prompt that the corresponding KV pair has already existed;}}  Else{  Insert the hash value corresponding to the Key;   Find the corresponding NVM device group;   Write the KV pair using the insert KV interface of NVM device;}

The pseudocode for the KV pair update operation in the group collaborative management based on BloomFilter is as follows.

  Collaborative_Updata (Key, Value)  Read the Key and Value of the KV pair;  Use Counting Bloomfilter to calculate the hash value of the Key;  If (There is a corresponding hash value) {    Find the corresponding NVM device group;   Call the lookup KV interface of the NVM device;    If (There is no corresponding Key) {      Return a prompt that there is no corresponding KV pair;}    Else{      Update the corresponding KV pair using the update KV interface of NVM device;}}  Else{   Return a prompt that there is no corresponding KV pair;}

The pseudocode for the KV pair get operation in the group collaborative management based on BloomFilter is as follows.

  Collaborative_Get(Key)  Read the Key corresponding to the KV pair to be read;  Use Counting Bloomfilter to calculate the hash value of the Key;   If (There is not corresponding hash value){   Return information that there is no corresponding KV pair;}  Else{   Find the corresponding NVM device group;   Call the lookup KV interface of the NVM device;    If (There is no corresponding KV pair in the NVM device){    Return a prompt that there is no corresponding KV pair;}   Else {      Obtain a corresponding KV pair by using the get KV interface of NVM device;  Return the KV pair to the KV application;} }

The pseudocode for the KV pair delete operation in the group collaborative management based on BloomFilter is as follows.

  Collaborative_Delete(Key)  Read the Key corresponding to the KV pair to be deleted;  Use Counting BloomFilter to calculate the hash value of the Key;   If (There is not corresponding hash value) {   Return information that there is no corresponding KV pair; }  Else{   Find the corresponding NVM device group;  Call the lookup KV interface of the NVM device;    If (There is no corresponding KV pair in the NVM device) {    Return a prompt that there is no corresponding KV pair;}   Else {     Call the delete KV interface of the NVM device to delete the corresponding KV pair;  Return the prompt to delete the KV pair successfully; } }

### 4.3. The Intragroup Optimization Based on Competition

The performance of NVM devices has instability due to wear leveling and garbage collection. The read/write speed of NVM devices will change with different amount of data stored and different period of lifetime. The group collaborative management based on BloomFilter cannot adapt to the different read/write speeds of NVM devices in the NVM array. We design an intragroup optimization based on competition. Each NVM device group contains a multiple of NVM devices, and a competition mechanism is used among these NVM devices to improve the efficiency of KV pair management. When an NVM device group receives a KV pair access request, it is simultaneously sent to all NVM devices in this NVM device group. These NVM devices simultaneously execute this access request and the result should be returned to the KV application when the first NVM device finishes it.

When using the intragroup optimization based on competition, the pseudocode for the KV pair competitive read operation is as follows.

  Competitive_Get (Key)  Read the value of Key corresponding to the KV pair to be read;  Call the lookup KV interface of all NVM devices in the same NVM device group;  If (an NVM device has found the corresponding Key) {  Return the corresponding KV pair to the KV application;   Cancel lookup operation of other NVM devices in the same NVM device group:}  Else  If (There are some NVM devices that do not complete the lookup operation)   continue waiting;   Else   Return information that there is no corresponding KV pair;

The pseudocode for the KV pair competitive insert operation in the intragroup optimization based on competition is as follows.

  Competitive_Insert (Key, Value)  Read the Key and Value of the KV pair;   Call the insert KV interface of all NVM devices in the same NVM device group;  If (an NVM device has completed the insert operation)  Return successful information;   Else   continue waiting; 

The pseudocode for the KV pair competitive update operation in the intragroup optimization based on competition is as follows.

  Competitive_Update (Key, Value)  Read the Key and Value of the KV pair;   Call the update KV interface of all NVM devices in the same NVM device group;  If (an NVM device has completed the update operation)  Return successful information;  Else   continue waiting; 

The pseudocode for the KV pair competitive delete operation in the intragroup optimization based on competition is as follows.

  Competitive_Delete (Key)  Read the Key corresponding to the KV pair to be deleted;   Call the delete KV interface of all NVM devices in the same NVM device group;  If (an NVM device has completed the delete operation)  Return successful information;   Else   continue waiting; 

Using intragroup optimization based on competition can mitigate the impact of NVM devices on the performance of KV store. By competing with other NVM devices in the same NVM device group, the access speed of KV pairs can be improved and the throughput of searching KV pairs can also be increased.

### 4.4. The Embedded KV Pairs Management Based on B+-Tree

In MKVS, the NVM device is only used to store the KV pairs. The embedded processor in the NVM device is used to establish an index of the KV key based on B+-Tree in each NVM device. There is an embedded KV engine in every NVM device, and it can manage the KV pairs using the embedded processor. A KV operation, such as lookup KV, get KV, update KV, insert KV, and delete KV, is implemented by adding an embedded KV engine to each NVM device. Then the NVM device can manage KV pairs autonomously. At the same time, the traditional byte or block interface of storage device is changed, and some new interfaces of KV operation such as lookup, get, insert, delete, and update of the KV pair are added to the NVM device. Then the result of the KV management operation can be conveniently obtained from the NVM device.

The embedded KV engine in each NVM device can be used to construct a distributed KV store in one host. It can be used to reduce the time and space overhead of KV pairs management, and has good scalability. It also can avoid the transfer a large amount of data between CPU and the NVM device when searching KV pairs. Then the interface of NVM device should not be the bottleneck of KV store. At the same time, by changing the interface of the NVM device and implementing the KV access interface, the I/O system software overhead can be effectively reduced and the efficiency of the KV application can be improved.

### 4.5. The Interface of MKVS

The current I/O system software stack needs to pass through multiple levels, such as cache, file system, and general block layer, to access KV pairs in the NVM device. It affects the read/write speed of KV store. Meanwhile, the conventional byte or block interface could not let the KV application to access KV pairs by using the embedded KV engine in NVM devices.

We short the I/O system software stack of MKVS for KV applications. As shown in Figure 2, the KV access interfaces are added to MKVS, and system calls corresponding to these KV access interfaces are added at the same time. Therefore, KV applications can skip several layers in the I/O system software stack, such as KV Store, file operating system call, file system, general block layer, etc., directly access KV management operations embedded in MKVS, and complete access and management functions required by KV application.

Therefore, the I/O system software stack for accessing the KV pair can be shortened and the efficiency of accessing and managing the KV pair can be improved. The KV management operation is encapsulated by the system call, which can ensure compatibility with existing applications and avoiding a lot of modifications to KV applications.

## 5. Evaluation

First, we implemented a prototype of the matrix KV storage system based on NVM devices (MKVS), and then used the Yahoo! Cloud System Benchmark (YCSB) to test and compare with the main in-memory KV store. YCSB includes a set of core workloads that define a basic benchmark. The Workloada in YCSB is a typical read–write mixed workload; the ratio of read and write requests is 50%. The Workloadb in YCSB is used to test the read performance and more than 95% access request are read operation.

### 5.1. The Prototype

Currently there are no commercial NVM devices. PMEM is a popular open source NVM device simulator. It is used to simulate NVM devices with default configuration. Then the group collaboration management module based on BloomFilter, intragroup optimization module based on competition, embedded KV management module based on B+-tree, and some new system calls are implemented to construct the prototype of MKVS. It contains two PMEM groups and there are two PMEM in each group.

The machine used for testing has two Intel E5 processors, 128GB of RAM, and a 120GB SAS port SSD hard drive. Centos7 is used as the operating system and the kernel version is 4.4.112. YCSB is used as a test tool. The five basic interfaces—read, insert, delete, update, and scan—were implemented for YCSB. Meanwhile, YCSB is run on local mode to avoid the impact of the network. There are two stages in the YCSB: Load and Run. The KV pairs should be inserted in KV store in Load stage and the access workload of KV pairs should be done in Run stage.

In order to effectively test and analyze MKVS, three prototypes are implemented named MKVS, MKVS-GEKV, and MKVS-EKV. MKVS is a full-scale prototype that includes the group collaboration management module based on Bloomfilter, intragroup optimization module based on competition, and embedded KV management module based on B+-tree. MKVS-GEKV is a prototype that lacks the intragroup optimization module based on competition. MKVS-EKV is a prototype of embedded KV management based on B+-tree. In addition, the three popular in-memory KV store were also used to test and compare: Redis, MongoDB, and Memcached. Meanwhile, the two KV stores based on PMEM, named PMEM-KV and PMEM-Redis, are also use to test.

### 5.2. Write Performance of KV Pairs

At first, 1000 KV pairs should be inserted into the KV store in the Load phase of YCSB to test throughput, time overhead, and average delay in a single-threaded mode. The results are shown in Table 1.

As can be found from Table 1, MKVS performs best efficiency of inserting KV pairs. Compared with MKVS-GEKV and MKVS-EKV, the throughput of inserting KV pairs has increased by 6% and 10%, respectively. Compared with Redis, MongoDB, and Memcached, the MKVS, MKVS-GEKV, and MKVS-EKV also can improve the significant throughput. MKVS-EKV has the lowest throughput among MKVS-GEKV and MKVS-EKV, but its throughput is increased by 2.5 times compared with MongoDB and 6% compared with Redis. While MKVS has the highest throughput, its throughput is increased by 2.8 times compared with MongoDB and is 16% higher than Redis. Meanwhile, compared to PMEM-KV and PMEM-Redis, MKVS, MKVS-GEKV, and MKVS-EKV can also improve the throughput of inserting KV pairs. MKVS’s throughput increased by 15% and 16% compared with PMEM-KV and PMEM-Redis, respectively. These results show that matrix KV storage system based on NVM can effectively improve the insert throughput of KV pairs. In addition, compared to Redis, MongoDB, Memcached, PMEM-KV, and PMEM-Redis, the average waiting time for inserting KV pairs of MKVS and MKVS-GEKV is reduced by 6% to 73%, which can effectively reduce the execution time of writing KV pairs.

Then, the number of threads in YCSB was changed to 2, 4, 8, and 16, respectively, and the throughput of inserting KV pair is tested. The results are shown in Table 2.

From Table 2, it can be found that the throughput of inserting KV pairs increases with the number of threads. MKVS’s throughput of inserting KV pairs is increased by ~9–23% compared with PMEM-KV and ~8–21% compared with PMEM-Redis. Except for MongoDB and Memcached, the throughput of other prototypes show an upward trend before the number of thread is low than 8, and then decreased when the number of thread increased from 8 to 16. MKVS’s throughput of inserting KV pairs decreased by a minimum of 4%, while PMEM-KV dropped by 15% at the maximum. These results show that MKVS can reduce the impact of NVM devices on the performance of KV storage systems, and also shows that storage devices and system software will become the bottleneck of the performance of KV storage system with the large number of threads.

### 5.3. Read-Intensive Workload

There are more than 95% read operations in Workloadb of YCSB. We used it to test the read performance of the prototype. The number of threads is set to 1, the number of KV pairs is 1000, and the number of operations is 1000. The results are shown in Table 3, where RAVL is the average read latency.

From the results in Table 3, it can be seen that MKVS, MKVS-GEKV, and MKVS-EKV can improve the throughput of reading KV pairs by 16.7%~6.5 times. The ratio of increase is higher than the write throughput. This is also because the read speed of NVM devices is higher than write speed, and the MKVS can reduce more time overhead of the KV storage system and utilize the I/O performance advantage of the NVM device. MKVS lose 2% read throughput compared to MKVS-GEKV, which is due to the fact that the KV pairs stored by MKVS are distributed among NVM devices in one group. However, the read throughput of MKVS increases by 9% compared with MKVS-EKV, which shows that intragroup optimization based on competition can effectively improve the read efficiency of KV storage system. In addition, MKVS can effectively reduce the average read latency of KV pairs; the average read latency is reduced by ~4–99.6% compared with Redis, MongoDB, Memcached, PMEM-KV, and PMEM-Redis.

Secondly, we change the number of threads to perform Workloadb. The number of threads is 1, 2, 4, 8, and 16. The number of access operations is 1000 and the number of KV pairs is 1000. The results are shown in Table 4.

Similar to the change of write throughput, the read throughput is increased obviously with increasing number of threads. Except MongoDB, the read throughput of other prototypes decreases when the number of read threads increases from 8 to 16. The change of MKVS is just 6%. At the same time, after the number of read threads is more than 1, the read throughput of MKVS is higher than that of MKVS-GEKV, which indicates that the intragroup optimization based on competition can adapt to the multiple read threads and effectively improve the read bandwidth of the KV storage system. In addition, when the number of read threads increases from 1 to 16, the read throughput of MKVS increases by 41% at the most, while MKVS-GEKV increased by only 5%. At the same time, PMEM-KV and PMEM-Redis only increases by 3% and 5%. These results demonstrate that the intragroup optimization based on competition can be well adapted to multithreaded reading workload and improve the read performance of KV storage system.

Then, the number of KV pairs changed to 5000, 10,000, and 2000, respectively, and the throughput of prototypes was tested with Workloadb. The number of access operations is 1000 and the number of threads is 1. The results are shown in Figure 3.

Changing the number of KV pairs allows focusing on testing the adaptability with different amounts of read and write data. Figure 3 shows the results on Workloadb. It can be seen that the throughput of MKVS, MKVS-GEKV, and MKVS-EKV is always higher than other prototypes with the change of KV pair number. The read throughput of MKVS is the highest when the number of KV pairs is 10,000; it is 53.1% higher than that of PMEM-KV and 9.4 times higher than Mongodb. This indicates that MKVS can improve the read speed of KV pairs compared with current KV storage systems. When the number of KV pairs increased from 5000 to 20,000, the read throughput of MKVS first increases and then decreases. At the same time, the read throughput of Mongodb and Memcached increases slowly, but the read throughput of MKVS-GEKV, MKVS-EKV, PMEM-KV, PMEM-Redis, and Redis decreases continuously. This shows that MKVS can reduce the management time overhead of KV storage system, and improve read throughput of KV pairs. It also shows that the management time overhead of Mongodb and Memcached is too large, so that the read speed of KV pair is much slower than NVM devices.

Finally, the number of access operations is changed to 5000 and 10,000, respectively, and the throughput of prototypes are tested with Workloadb. The number of KV pairs is 1000 and the number of threads is 1. The results are shown in Figure 4.

From the results shown in Figure 4, the read throughput of MKVS, MKVS-GEKV, and MKVS-EKV is always higher than other prototypes with the change of the access operation number. When the number of access operations is 5000, MKVS can increase the read throughput by 0.19~11.7 times compared with Redis, MongoDB, Memcached, PMEM-KV, and PMEM-Redis. When the number of access operations is 10,000, it is 30%~18.6 times. The read throughput improvement of MKVS increases with the number of access operations. At the same time, when the number of access operations increases from 5000 to 10,000, MKVS has highest ratio between the increased read throughput and the number of access operations, its value is 6.4. Meanwhile, Memcached reaches 0.1 at the lowest. These results indicate that MKVS has high scalability when the number of access operations is changed.

### 5.4. Mixed Workload

There are many update requests in Workloada, the ratio of read and write requests is 50%. It is a typical read–write mixed workload. First, Workloada is used to test the performance of prototypes in single thread mode. The number of KV pairs is 1000 and the number of operations is 1000. The results are shown in Table 5, where the RAVL is average read latency and UAVL is average write latency.

From Table 5, it can be found that MKVS can effectively improve the read–write throughput compared with other prototypes. The largest throughput improvement is increased by 5.98 times compared with MongoDB, and the smallest throughput improvement is increased by 14.8% compared with PMEM-Redis. The read–write throughput of MKVS-GEKV and MKVS-EKV is always higher than other prototypes, except MKVS. At the same time, MKVS can reduce the read and write latency compared with Redis, MongoDB, Memcached, PMEM-KV, and PMEM-Redis. The read latency is reduced by ~2.9–99.7% and the write latency is reduced by ~12.8–77.2%. These results show that MKVS can effectively improve the response speed of reading and writing KV pairs.

Then we change the number of threads to perform Workloada. The number of threads is 1, 2, 4, 8, and 16, respectively. The number of KV pairs is 1000 and the number of operations is 1000. The results are shown in Table 6.

From the results of Table 6, it can be found that MKVS has the highest read–write throughput when executing Workloada with different number of threads. At the same time, the throughput of MKVS-GEKV and MKVS-EKV is always higher than other prototypes. The read–write throughput of all prototype systems decreased, when the number of threads increased from 8 to 16. The most obvious degradation is Redis by 12.6%, while MKVS is only 5.7%. This shows that MKVS has good stability of read–write throughput.

Thirdly, the number of KV pairs is changed to 5000, 10,000, and 20,000, respectively, and the throughput of prototypes are tested with Workloada. The number of access requests is 1000 and the number of threads is 1. The results are shown in Figure 5.

As can be seen from Figure 5, similar to the results with Workloadb, the throughput of MKVS, MKVS-GEKV, and MKVS-EKV is always higher than other KV storage systems. When the number of KV pairs is 10,000, MKVS has the highest read–write throughput. Meanwhile, its throughput is 34.8% higher than PMEM-Redis and 2.9 times higher than Memcached. This indicates that MKVS has the read and write speed advantage of KV pairs compared with current KV storage systems. Unlike the read throughput with Workloadb, when the number of KV pairs increases from 5000 to 20,000, the read–write throughput of all prototype systems first increases and then decreases.

Finally, the number of access operations is changed to 5000 and 10,000, respectively, and the throughput of prototypes are tested with Workloada. The number of KV pairs is 1000 and the number of threads is 1. The results are shown in Figure 6.

From the results shown in Figure 6, the read throughput of MKVS, MKVS-GEKV, and MKVS-EKV is always higher than other prototype systems with the change of access operations number. When the number of access operations is 5000, MKVS can increase the read–write KV throughput by 0.34~7.8 times compared to Redis, MongoDB, Memcached, PMEM-KV, and PMEM-Redis. It is 0.22~11.4 times when the number of access operations is 10,000. At the same time, when the number of access operations increases from 5000 to 10,000, MKVS has highest ratio between the increased read throughput and the number of access operations, its value is 4.14. Meanwhile, Memcached reaches 0.14 at the lowest. These results indicate that MKVS is suited to handle a large number of access operations and improve the read–write throughput of KV pairs.

## 6. Conclusions

We analyzed the characteristics of NVM devices and demands of KV application to design a group collaboration management based on Bloomfilter, intragroup optimization based on competition, embedded KV management based on B+-tree, and a new interface for the KV storage system. Then the matrix KV storage system based on NVM devices can be constructed. The embedded processor in the NVM device and CPU can be comprehensively utilized to improve the efficiency of massive KV pair management. It also can improve the efficiency of KV applications execution. The prototype is implemented named MKVS to test with YCSB. The results show that PMEKV has the advantage of throughput, latency and adaptability compared with current in-memory KV stores.

Now we just use the Bloomfilter and B+-tree, they lack the optimization of efficiency and concurrency to the NVM devices. In the future, we should study the distributed and index algorithm for the matrix KV storage system.

## Figures and Tables

**Figure 1 micromachines-10-00346-f001:**
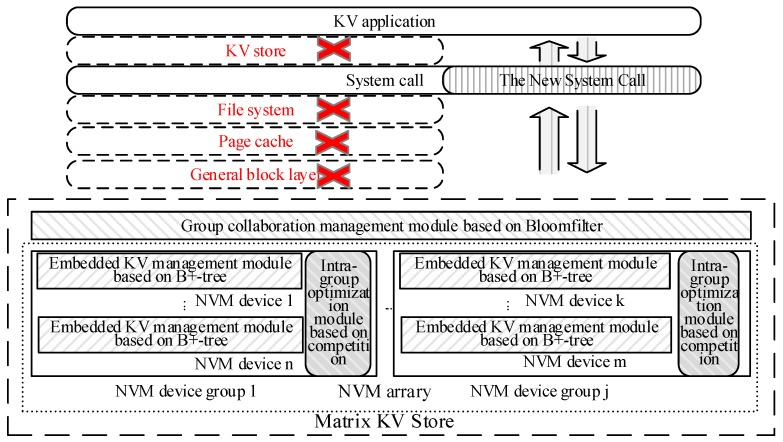
The structure of MKVS.

**Figure 2 micromachines-10-00346-f002:**
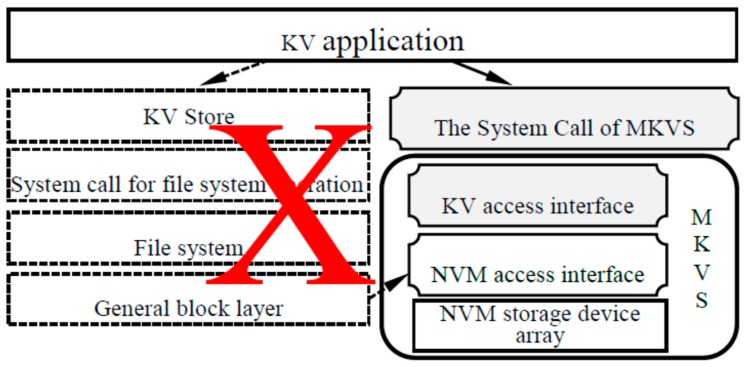
Schematic diagram of the MKVS access interface.

**Figure 3 micromachines-10-00346-f003:**
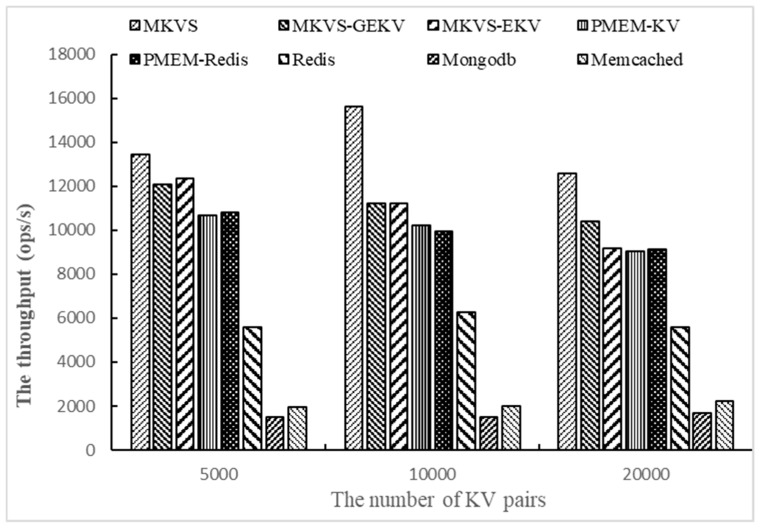
The throughput of changing the number of KV pairs with Workloadb.

**Figure 4 micromachines-10-00346-f004:**
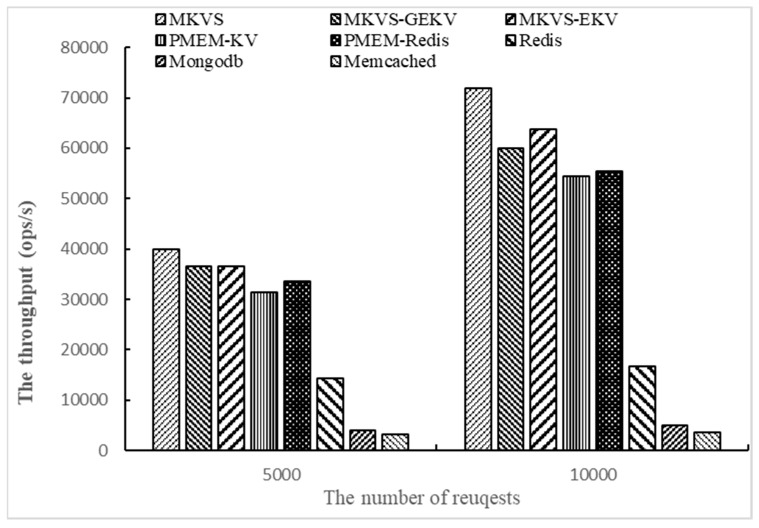
The throughput of changing the number of access operations with Workloadb.

**Figure 5 micromachines-10-00346-f005:**
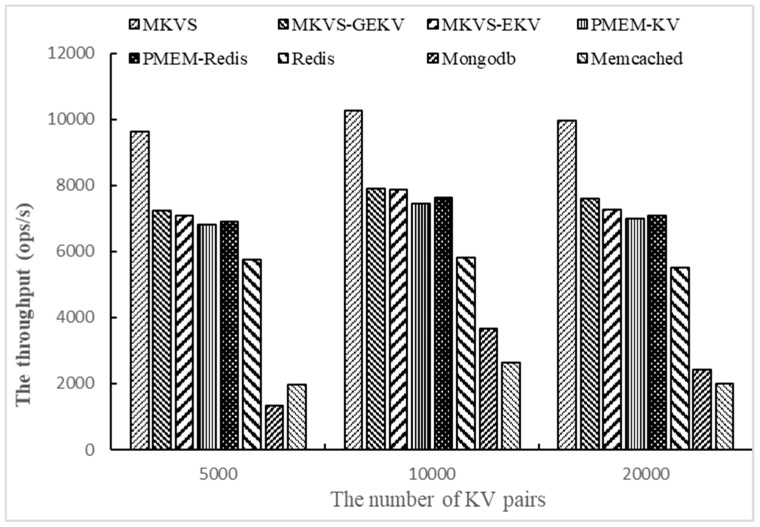
The throughput of changing the number of KV pairs with Workloada.

**Figure 6 micromachines-10-00346-f006:**
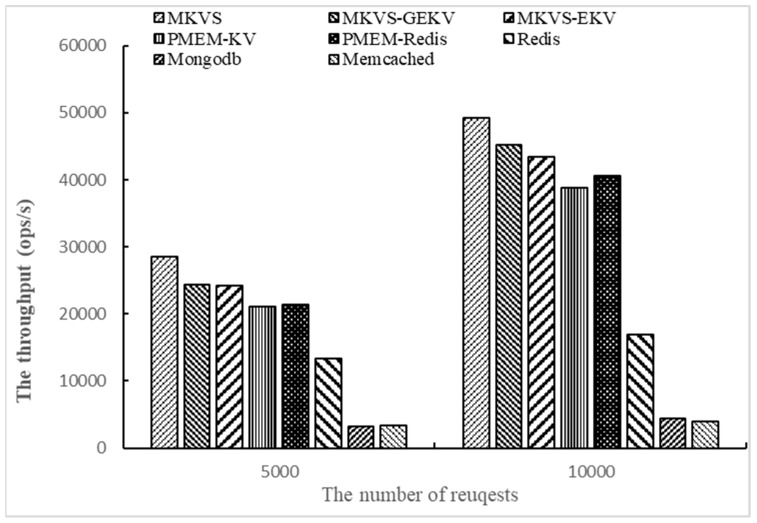
The throughput of changing the number of access operations with Workloada.

**Table 1 micromachines-10-00346-t001:** The results of inserting 1000 KV pairs.

	MKVS	MKVS-GEKV	MKVS-EKV	PMEM-KV	PMEM-Redis	Redis	MongoDB	Memcached
Throughput (ops/s)	5226	4933	4765	4539	4603	4500	1364	1818
Runtime (ms)	196	201	215	220	218	222	733	550
Average Latency (ms)	135	139	176	148	144	150	503	417

**Table 2 micromachines-10-00346-t002:** The results of inserting KV pairs with multithreads.

Throughput (ops/s)	MKVS	MKVS-GEKV	MKVS-EKV	PMEM-KV	PMEM-Redis	Redis	MongoDB	Memcached
**2 threads**	6580	6320	5451	5397	5702	5649	1808	1960
**4 threads**	7301	6994	6113	6064	6057	6037	2409	2481
**8 threads**	7526	7313	7143	6915	6995	6944	2493	2174
**16 threads**	7221	6759	6172	5873	6112	6012	2770	2247

**Table 3 micromachines-10-00346-t003:** The performance with read-intensive workload.

	MKVS	MKVS-GEKV	MKVS-EKV	PMEM-KV	PMEM-Redis	Redis	MongoDB	Memcached
**Throughput (ops/s)**	11,805	12,083	10,869	10,154	10,358	6451	1615	1700
**Runtime (ms)**	80	75	92	101	98	155	619	588
**RAVL (ms)**	1.62	1.55	1.67	1.69	1.68	80	359	436

**Table 4 micromachines-10-00346-t004:** The results of multithreads with Workloadb.

Throughput (ops/s)	MKVS	MKVS-GEKV	MKVS-EKV	PMEM-KV	PMEM-Redis	Redis	MongoDB	Memcached
**Single thread**	11,805	12,083	10,869	10,154	10,358	6451	1615	1700
**2 thread**	14,467	14,322	12,157	10,538	11,254	7936	1956	1488
**4 thread**	15,950	15,353	12,987	11,293	12,036	9708	2403	2695
**8 thread**	16,601	16,029	11,363	10,481	10,879	9433	2439	2525
**16 thread**	15,477	14,728	10,866	9588	10,258	8547	2652	2004

**Table 5 micromachines-10-00346-t005:** The performance in single thread mode with Workloada.

	MKVS	MKVS-GEKV	MKVS-EKV	PMEM-KV	PMEM-Redis	Redis	MongoDB	Memcached
**Throughput (ops/s)**	8624	8504	7863	7455	7514	5714	1262	1890
**Runtime (ms)**	110	112	133	138	136	175	792	529
**RAVL (ms)**	1.34	1.33	1.36	1.38	1.38	101	411	386
**UAVL (ms)**	150	158	167	177	172	85	658	404

**Table 6 micromachines-10-00346-t006:** The throughput of Workloada with different threads.

Throughput (ops/s)	MKVS	MKVS-GEKV	MKVS-EKV	PMEM-KV	PMEM-Redis	Redis	MongoDB	Memcached
**2 threads**	10,854	9851	9233	8530	8823	8124	1893	2145
**4 threads**	11,953	11,524	9708	9434	9673	9618	2183	2906
**8 threads**	12,833	12,571	11,069	9503	11,014	10,894	2717	2597
**16 threads**	12,097	11,684	10,526	9148	9749	9523	2409	2369

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
