# Peer review of "The Matrix KV Storage System Based on NVM Devices"

_micromachines, 2019, doi:10.3390/mi10050346_

Round 1

Reviewer 1 Report

-Acronyms should be defined in the abstract and again the first time used in the paper.

(i.e.YCSB line 329 etc).

-it would be helpful mentioning  the workloada, and the workloadb along with their importance  in the beginning of section 5.Evaluation (line after 329)

Author Response

Point 1: Acronyms should be defined in the abstract and again the first time used in the paper. (i.e.YCSB line 329 etc).

Response 1: I have modified the manuscript to add terminology definition for each abbreviation. For example, Yahoo! Cloud System Benchmark (YCSB).

Point 2: it would be helpful mentioning the workloada, and the workloadb along with their importance in the beginning of section 5.Evaluation (line after 329).

Response 2: We add the introduction of Workloada and Workloadb, and give the reasons for using them.

Reviewer 2 Report

The manuscript (micromachines-504609) shows an interesting results of matrix KV storage system based on NVM devices design. Although authors show a quite comprehensive analysis on the matrix comparison, some comments still need authors to address before further confirmation, list following:

Authors should review the whole manuscript and add the terminology definition for each abbreviation (with Abstract).

Since authors mentioned in the introduction "Therefore, how to design new KV storage systems according to characteristics of NVM devices and KV application is an important factor in improving performance of KV application.", can authors provide the comments and KV storage system guidance design requirement for different NVM devices (RRAM, PCM, SXP, MRAM, FeRAM)? For example, for RRAM, what kind of KV storage systems are requirement according to characteristics of RRAM devices? That would be quite useful information for later emerging NVM development. 

Author Response

Point 1: Authors should review the whole manuscript and add the terminology definition for each abbreviation (with Abstract). Acronyms should be defined in the abstract and again the first time used in the paper. (i.e.YCSB line 329 etc).

Response 1: I have modified the manuscript to add terminology definition for each abbreviation. For example, Yahoo! Cloud System Benchmark (YCSB) and NVM (NVM devices).

Point 2: Since authors mentioned in the introduction "Therefore, how to design new KV storage systems according to characteristics of NVM devices and KV application is an important factor in improving performance of KV application.", can authors provide the comments and KV storage system guidance design requirement for different NVM devices (RRAM, PCM, SXP, MRAM, FeRAM)? For example, for RRAM, what kind of KV storage systems are requirement according to characteristics of RRAM devices? That would be quite useful information for later emerging NVM development.

Response 2: We design the MKVS based on some characteristics of NVM devices such as byte-addressable, in-place update, longer endurance as well as higher read and write speed compared with Flash. Which was given in section 1 (Introduction). Different from the SSD based on Flash, the embedded processor need not to deal with a lot of extra control of NVM devices such as GC and WL. Therefore, the embedded processor can be used to manage KV pair and provide a KV interface for KV application. However, some NVM still have the limitation of endurance, which affects the efficiency of MKVS. The NVM with long writing lifetime is more suitable for MKVS. Meanwhile, the interface of NVM devices is also a challenge. In this work, we design the MKVS for the NVM device with PCIe interface. If the NVM device is with DIMM interface, something maybe changed. The inter structure of NVM is also a problem, 3D Xpoint maybe a useful way yet it will affect the management method of data in NVM devices.
